# The Combined Effects of the Membrane and Flow Channel Development on the Performance and Energy Footprint of Oil/Water Emulsion Filtration

**DOI:** 10.3390/membranes12111153

**Published:** 2022-11-16

**Authors:** Nafiu Umar Barambu, Muhammad Roil Bilad, Norazanita Shamsuddin, Shafirah Samsuri, Nik Abdul Hadi Md Nordin, Nasrul Arahman

**Affiliations:** 1Chemical Engineering Department, Universiti Teknologi PETRONAS, Bandar Seri Iskandar 32610, Perak, Malaysia; 2Faculty of Integrated Technologies, Universiti Brunei Darussalam, Jalan Tungku Link, Gadong BE1410, Brunei; 3Department of Chemical Engineering, Universitas Syiah Kuala, Jl. Syeh A. Rauf, No. 7, Banda Aceh 23111, Indonesia

**Keywords:** crossflow membrane filtration, membrane surface development, energy saving, sustainable engineering, membrane fouling, oil/water emulsion

## Abstract

Membrane filtration is a promising technology for oil/water emulsion filtration due to its excellent removal efficiency of microdroplets of oil in water. However, its performance is highly limited due to the fouling-prone nature of oil droplets on hydrophobic membranes. Membrane filtration typically suffers from a low flux and high pumping energy. This study reports a combined approach to tackling the membrane fouling challenge in oil/water emulsion filtration via a membrane and a flow channel development. Two polysulfone (PSF)-based lab-made membranes, namely PSF- PSF-Nonsolvent induced phase separation (NIPS) and PSF-Vapor-induced phase separation (VIPS), were selected, and the flow channel was modified into a wavy path. They were assessed for the filtration of a synthetic oil/water emulsion. The results showed that the combined membrane and flow channel developments enhanced the clean water permeability with a combined increment of 105%, of which 34% was attributed to the increased effective filtration area due to the wavy flow channel. When evaluated for the filtration of an oil/water emulsion, a 355% permeability increment was achieved from 43 for the PSF-NIPS in the straight flow channel to 198 L m^−2^ h^−1^ bar^−1^ for the PSF-VIPS in the wavy flow channel. This remarkable performance increment was achieved thanks to the antifouling attribute of the developed membrane and enhanced local mixing by the wavy flow channel to limit the membrane fouling. The increase in the filtration performance was translated into up to 78.4% (0.00133 vs. 0.00615 kWh m^−3^) lower in pumping energy. The overall findings demonstrate a significant improvement by adopting multi-pronged approaches in tackling the challenge of membrane fouling for oil/water emulsion filtration, suggesting the potential of this approach to be applied for other feeds.

## 1. Introduction

The membrane-based process has long been recognized as a reliable technology for process separation. It has been widely adopted as the best option for decentralizing water and wastewater treatment [1,2,3]. It has also been widely explored to treat oily wastewater, including oil/water emulsions [4]. However, the economic advantages of the technology are very sensitive to the energy input that drives the filtration. In crossflow membrane filtration, the energy is consumed mainly for feed pumping, and the specific energy consumption is a function of the applied pressure and achievable permeability. Like other membrane-based processes, the system throughput is dictated by the ability to manage membrane fouling, which is typically managed in an oil/water emulsion filtration by minimizing the interaction between the oil droplets and the membrane surface [5,6,7].

For surfactant stabilized oil/water emulsion, like the water produced in the oil and gas industries, polymeric membranes offer high separation efficiency, mainly when containing oil droplets smaller than 20 µm. A polymeric membrane can be easily prepared by using the conventional phase inversion. Most commercial polymeric membranes are prepared using water as the non-solvent, requiring the main polymer to be hydrophobic, resulting in a hydrophobic membrane [8,9,10]. The hydrophobic attribute of the polymeric membrane makes it vulnerable to fouling when handling oil droplets in the oil/water emulsion feed because of the inherent oil affinity of the surface [11,12,13].

Many strategies have been explored to restrict oil droplet/membrane interactions in oil/water emulsion filtration systems using membranes. They include membrane material development and membrane module engineering [14,15,16] or a combination thereof [17]. Those approaches have resulted in remarkable improvements in hydraulic performance. Most membrane material developments have focused on imposing hydrophilic properties on the membrane surface, increasing filtration fluxes to >90 L m^−2^ h^−1^ [16]. Some materials offer remarkably high filtration flux, reaching ≈12,000 L m^−2^ h^−1^ [18]. Most of those ultra-high permeable membranes were evaluated rapidly under extremely low-pressure filtration systems that ignored membrane compaction and irreversible fouling (in multi cycles and long-term tests) [19], which might be incompatible with being used as a basis for full-scale installation. A more conservative improvement in oil/water emulsion flux was demonstrated in our recent work by employing a simple vapor-induced phase separation (VIPS) technique [20]. The flux increased from 9.0 to 27.8 L m^−2^ h^−1^ under a feed gauge pressure of 0.2 bar over 10 h of filtration.

Another strategy to enhance the performance of oil/water emulsion filtration is by imposing a surface groove to form patterns on the membrane surface instead of the conventional flat surface. The surface pattern is aimed to promote the feed turbulence flow and to generate fluid eddies that restrict the interplay of oil droplets with the membrane surface [21,22]. Moreover, fluid eddies also reduce the boundary layer resistance and bestow a self-cleaning effect upon the membrane, thereby enhancing filtration performance and, consequently, improving energy saving [23,24]. Unfortunately, creating a pattern on the membrane surface often requires pre-optimization of the fabrication method, resulting in a membrane material with an entirely new set of properties [25,26,27]. This approach also involves sophisticated fabrication methods [25,26,28,29]. Recently, we reported a new approach to promoting local mixing on the membrane surface by developing a wavy flow channel [30]. It effectively enhanced an oil/water emulsion flux by 58% to 43.8 L m^−2^ h^−1^.

Besides boosting the filtration throughput, the energy input affects the sustainability of crossflow membrane filtration. A filtration system with a membrane material with high permeability and a low fouling propensity led to low energy consumption. For example, optimizing the ΔP and the membrane in a conventional crossflow filtration of oil/water emulsion could reduce the pumping energy input by 66% [31]. For other feeds, intensifying the filtration via the membrane surface corrugation contributed to an 88% energy saving [32]. In another report, utilizing the disk rotation in a conventional rotating biological contactor to control the membrane fouling of a membrane placed between the disks offered a ~72% energy saving [33]. For those reports, the energy-saving performance was attributed to the system’s enhanced feed turbulence flow and the applied membrane’s antifouling properties, demonstrating their importance in enhancing the hydraulic performance and lowering the energy footprint of a filtration system.

This study evaluated the synergistic effect of membrane and module developments for oil/water filtration. Each approach was investigated individually in our previous works [20,30]. Their combined contribution to enhancing the hydraulic performance and lowering the energy input is reported for the first time in this study. Two flat-sheet membranes were made in a laboratory and tested under a conventional straight flow channel and a newly developed wavy flow channel. The hydraulic performance of the membrane was then evaluated. Finally, their energy consumption in a hypothetical full-scale module for oil/water emulsion treatment was compared. This methodology allowed us to distinguish between the advantages gained from membrane and module developments independently.

## 2. Materials and Methods

### 2.1. Preparation of the Oil/Water Emulsion and Membrane Samples

The synthetic oil/water emulsion sample used as the filtration feed was prepared by mixing an actual crude oil sample (obtained from a production well in Southeast Asia) with distilled water and a surfactant, in the form of sodium dodecyl sulfate (SDS, 98% purity, Sigma Aldrich, St. Louis, MO, USA), to stabilize the emulsion. The feed was set to have 1000 ppm of crude oil concentration. The emulsion was formed by stirring the mixture of water, crude oil, and surfactant at 350 rpm for 24 h. It was visually stable, without any sign of phase separation or floatation of the oil layer on the surface. The intensity distribution of oil droplets in water was multimodal, with dominant sizes of 0.25, 0.01, and 4.0 µm.

Two lab-made membranes were used to evaluate the impact of membrane development on hydraulic performance and pumping energy. Our earlier work [14] reported that they were pre-developed using the VIPS method. The main polymer for membrane fabrication was polysulfone (PSF, MW of 78 kDa, Sigma Aldrich), dissolved in dimethylacetamide (DMAc, 99.8% purity, Sigma Aldrich) as the solvent, in addition to an additive comprising both lithium chloride (LiCl, MW of 42.38 g/mol, ACROS Organics, Geel, Belgium) and polyethylene glycol (PEG, MW of 10 kDa, Sigma Aldrich). The dope solutions were prepared at a fixed composition for PSF, DMAc, PEG, and LiCl of 18 wt%, 80.9 wt%, 1 wt%, and 0.1 wt%, respectively. The homogeneous dope solution was cast onto a nonwoven textile as the support (Novatexx 2413, Freudenberg Filtration Technologies, Weinheim, Germany) at a wet casting thickness of 220 µm, followed by immersion in distilled water as the non-solvent. The difference between the two membranes was the time gap between the casting and immersion. The first sample, PSF-Nonsolvent induced phase separation (NIPS) was prepared with no time gap, while the second sample was prepared with a time gap of 60 s (PSF-VIPS) under the ambient temperature and relative humidity of 22 °C and 70%, respectively.

### 2.2. Filtration Test

The filtration tests were run in the custom-made crossflow filtration setup illustrated in Figure 1A. All filtrations were done under a constant feed gauge pressure of 0.2 bar, a constant feed crossflow velocity of 5.5 cm/s, and a full-recirculation system. The filtration was driven by the △*P* generated by the feed pump that was also used to circulate the oil/water emulsion feed. The collected permeate was returned every 10 min to maintain the feed condition. The oil concentrations of the feed and permeate samples were determined using a UV-VIS spectrometer (Shimadzu UV-2600, Kyoto, Japan) at a wavelength of 227 nm.

Two types of filtration cells were applied in this study. The first was the standard cell with a straight flow channel, mounted with a membrane sample with an effective filtration area of 37 cm^2^. The second was a modified cell with a wavy flow channel (Figure 1B,C). The projected width and length of this cell were similar to the straight flow channel. Due to its wavy design, the cell was mounted by a 34% longer membrane sheet, corresponding to an effective filtration area of 49.6 cm^2^. The flow channel’s amplitude, wavelength, and flow depth were set at 5, 20, and 2 mm, respectively.

The filtration cells were used to measure the clean water permeability (CWP, *L* in L m^−2^ h^−1^ bar^−1^) of the membranes and their filtration performance for the oil/water emulsion feed. First, the CWP was measured by the filtration of distilled water. The membrane was first compacted for 60 min, then continued for 30 min, where the permeate was collected every 10 min. The volume data were used to evaluate the CWP using Equation (1). Subsequently, the feed was exchanged with the oil/water emulsion, and the filtration was extended for 90 min. During the filtration of the oil/water emulsion, the permeate volume was recorded every 10 min. After the volume measurement, it was returned to the feed tank. The permeate volume data were then used to calculate the oil/water emulsion permeability using Equation (1). Next, the feed was exchanged with distilled water, and the filtration was continued for 30 min to evaluate the fouling reversibility of the filtration system. The CWP data for the final 10 min were then recorded. Ninety minutes of oil/water emulsion filtration, followed by 30 min with distilled water, comprised one filtration cycle. The filtration tests consisted of five cycles.
(1)$$L=\frac{\Delta V}{A\;\Delta t\;\Delta P}$$
where ΔV represents the permeate volume (L) obtained for each filtration cycle for a period (Δt) of 10 min under a constant transmembrane pressure (ΔP) of 0.2 bar in a filtration cell with an effective membrane area of A (m^2^).

### 2.3. Estimation of Pumping Energy Consumption

The feed pump is responsible for most of the energy consumption of a crossflow membrane filtration system. This study represented the pumping energy by the specific energy consumption (*E*, KWh m^−3^) estimated using Equation (2). The ṁ in the equation denotes the mass rate (kg s^−1^) expressed by Equation (3), while $W_{p}$ denotes the work of the feed pump (J kg^−1^) expressed by Equation (4).

The energy estimation was done for a hypothetical full-scale plate and frame panel with a length of 2 m, a width of 1 m, and the flow channel width or the gap between two adjacent panels of 2 mm. The system was operated under similar filtration conditions to those applied in the experiments, with a crossflow velocity and ΔP of 5.5 cm/s and 0.2 bar, respectively. The pressure drop data along the module were obtained from the lab-scale measurement, and the permeability values were obtained from the experimental data.
(2)$$E=\frac{ṁ\;W_{p}}{V_{P}}$$
(3)ṁ=ρVwL
(4)$$W_{p\;}=\frac{\frac{P}{\rho }+\frac{V^{2}}{2}+F}{3,600,000}$$
where ρ denotes the water density (kg/m^3^), 𝒱 represents the operating linear feed velocity (m/s), w is the membrane width (m), L is the feed flow depth (m), P is the inlet pressure (Pa), and F is the flow channel frictional loss (m^2^/s^2^).

## 3. Results and Discussion

### 3.1. Membrane Characteristics

Two membranes were assessed to demonstrate the impact of the membrane properties on the crossflow filtration energy consumption. Their development and characterization were reported in our earlier work [20]. The PSP-NIPS membrane had a thickness of 218.3 ± 1.3 µm with a surface water contact angle of 70.3 ± 0.6°. On the other hand, the PSF-VIPS had a thickness of 235.7 ± 1.7 µm with a surface water contact angle of 57.7 ± 0.6°. Membrane fabrication using VIPS allowed the formation of an immobile layer on the cast polymer film during exposure to humid air. This layer hindered the outflow mobility of the hydrophilic additive toward the non-solvent during the phase inversion process. This phenomenon led to the formation of more hydrophilic surface chemistry. Conversely, for the NIPS method, the hydrophilic additive mostly leached out to the non-solvent during the phase inversion. In a phase-inverted membrane, the top dense layer is mainly responsible for filtration resistance, and the overall membrane thickness is less significant. The PSF-VIPS and the PSF-NIPS exhibited pore size and porosity of 0.032 µm, 73.2 ± 0.2%, and 0.126 µm and 58.1 ± 0.2%, respectively. The PSF/NIPS was a plain membrane used as a reference, while the PSF-VIPS was a modified membrane based on vapor-induced phase separation. When evaluated in the common straight flow channel, the latter had a substantial advantage. This study explored this advantage when applied in the wavy flow channel regarding the hydraulic performance and energy footprint.

### 3.2. Effect of the Membrane and Module Flow Channel on Clean Water Permeability

The membrane properties and module configuration can affect the module performance in membrane filtration [34,35]. The term “packing density” is based on the possibility of packing a specific membrane area in a particular module volume. Figure 2 compares the clean water permeability of the two membranes under the flat and wavy flow channels, showing clear advantages of the modified membrane under the wavy flow channel. The CWP of the PSF-NIPS and the PSF-VIPS were 329 ± 8 L m^−2^ h^−1^ bar^−1^ and 502 ± 9 L m^−2^ h^−1^ bar^−1^, respectively, corresponding to a 52.7% increment. When the modified membrane (PSF-VIPS) was evaluated under the wavy flow channel, the permeability was 673 ± 8 L m^−2^ h^−1^ bar^−1^, yielding an advantage of 105% over the PSF-NIPS. The advantage of PSF-VIPS over PSF-NIPS could be attributed to the improved properties of the membranes (Section 3.1) gained from the vapor-induced phase separation fabrication discussed in our earlier work [20].

On the other hand, the 34% advantage of the wavy flow channel was due to its advantage in increasing the module packing density (or effective filtration area), as also reported by others [23,29,36,37]. Due to its wavy flow trajectory, an additional 34% of membrane length could be packed. Since there was no fouling when treating clean water, the intrinsic clean water permeability of the flat and wavy channels was insignificant.

When treating the clean water, the hydraulic filtration performance was not affected by fouling. Hence, Figure 3 shows the full potential of the combined approach to membrane and module development. The advantage of the membrane in the two flow channels was determined accurately, because it was evaluated from the same original sheet. The simple flow channel modification allowed the use of a traditional flat-sheet membrane, which underwent re-optimization in many surface-patterned membrane systems [27,29,38]. As shown in Figure 2, clean water permeability yielded almost no advantage for either the flat or wavy (A) flow channels, with corresponding *p*-values of 0.6316 and 0.6464 from ANOVA, respectively. In an earlier study, surface patterns on membranes increased clean water permeability by 10–22% thanks to the pre-optimized membrane fabrication parameters, including the additional effective filtration area via a 3D support [39,40].

The additional effective area advantage offered by the wavy flow channel might seem indistinct. One could argue that such an advantage can be achieved by a traditional plate-and-frame module with a larger effective membrane area and still using the same membrane. This is valid, but it ignores a few advantages of the wavy flow channel system. The material used for membrane production was reduced, since a low thickness could be maintained without the need to provide space for developing surface patterns. The high packing density also led to a smaller overall module footprint, and hence a lower construction cost for the filtration tank [41].

### 3.3. Filtration of Oil/Water Emulsion

The combined advantages of both the membrane material and flow channel development on the permeability of the oil/water emulsion membrane filtration are demonstrated in Figure 3. It shows the performance of five filtration cycles. Each cycle comprised 90 min of oil/water filtration and 30 min water flushing. The PSF-VIPS membrane performed much better than the PSF-NIPS. Both membranes excelled when employed in the wavy flow channel compared to the straight flow channel. Comparing the permeability data from the fifth cycle, the overall advantage of combined membrane and flow channel development was 355%. The permeability of the straight PSF-NIPS was 43.4 L m^−2^ h^−1^ bar^−1^, much lower than the PSF-VIPS in the wavy flow channel, i.e., 197.5 L m^−2^ h ^−1^ bar^−1^.

The significant increment of oil/water emulsion permeability of the PSF-VIPS in the wavy flow channel could not only be attributed to the improved membrane properties (Section 3.1) and enhanced effective filtration area (34%, Figure 2), but also to the improved hydrodynamics offered by the wavy flow channel. The combined advantages of clean water permeability due to the improved membrane properties and membrane area were 105%. The advantages increased to 355% when used for filtration oil/water emulsion, in which membrane fouling played a crucial role in diminishing the filtration performance. The antifouling performance can be attributed to the combined effect of the antifouling attribute of the PSF-VIPS relative to the PSF-NIPS and the improved flow properties due to the wavy flow channel. Of the 355% permeability advantage, 241% could be attributed to the enhanced fluid dynamics in the wavy flow channel and the antifouling property of the PSF-VIPS.

Moreover, the fluid eddies generated in the wavy flow also increased foulant removal by driving oil droplets away from the membrane surface or leaving them insufficient time to interact with it [27,42]. The wavy patterns also bestowed axial flow resistance upon the membrane surface, enhancing the surface shear stress and thus restricting the oil droplet adsorption [23,29,43]. Our earlier reports [23] demonstrated that the membrane development contributed to a 66% energy saving by enhancing the oil/water permeability. As demonstrated by a reduced contact angle, membrane development via the VIPS fabrication method resulted in the membrane being more hydrophilic. A similar finding was reported for polyvinylidene difluoride using PEG as the additive [44]. A membrane surface with low hydrophilicity could hinder the formation of an oil layer on the membrane surface and facilitate water transport [45].

### 3.4. Reduction in Energy Consumption

Figure 4 demonstrates the contribution of the wavy flow channel and membrane development in lowering the specific pumping energy during oil/water emulsion filtration. The filtration using the wavy flow channel consumed less energy than the one using the straight flow channel across all filtration cycles. Even for the first filtration cycle, the wavy PSF-NIPS and PSF-VIPS membrane consumed a pumping energy of just 0.016 kWh/m^3^ and 0.008 kWh m^−3,^ respectively. In contrast, their flat counterparts consumed up to 0.022 kWh m^−3^ and 0.011 kWh m^−3^. Those decreases corresponded to a ~27% reduction, which is highly significant, considering the sensitivity of the energy footprint in engineering decisions. The wavy flow channel’s energy-saving gradually increased across the filtration cycles. For the second cycle, an energy-saving of up to ~28% was recorded; this increased to ~30% in the fourth cycle. When comparing the data from the fifth cycle, the energy savings attributed to the wavy flow channel were 33% for both PSF-NIPS and PSF-VIPS. This increasing trend can be attributed to the increasing influence of the hydrodynamics in limiting the filtration resistance caused by the boundary layer. The wavy membranes made the boundary layer thinner thanks to the fluid eddies induced by the swirling flow patterns, as demonstrated earlier by the fluid dynamics simulation [30].

Figure 4 suggests the advantages of multi-pronged approaches in reducing the pumping energy in a crossflow membrane filtration system, i.e., the highest energy consumption contributor. Comparing the baseline/reference of PSF-NIPS in straight flow configuration to PSF-VIPS in the wavy flow channel configuration, a staggering energy reduction of 78.2% was achieved. These findings will pave the way for future approaches to addressing the high energy footprint of membrane filtration.

## 4. Conclusions

The findings of this study demonstrate the effectiveness of multi-pronged approaches in enhancing the filtration performance and reducing the pumping energy of a crossflow membrane filtration system. The proposed membrane and flow channel enhanced the clean water permeability by 105% (329 vs. 673 L m^−2^ h ^−1^ bar^−1^). The wavy flow channel allowed additional packing of a 34% higher membrane area, thereby improving module productivity. Meanwhile, membrane development further enhanced the permeability by 52.7%. The intrinsic advantage in clean water permeability was further elevated when treating fouling-prone feed in the form of oil/water feed. The combined advantage of the flow channel and membrane developments resulted in a 355% permeability enhancement (from 43 to 198 L m^−2^ h ^−1^ bar^−1^). Finally, the hydraulic performance reduced the pumping energy by 78.2% (0.00133 vs. 0.00615 kWh m^−3^) due to the combined effect of membrane and flow channel development.

## Figures and Tables

**Figure 1 membranes-12-01153-f001:**
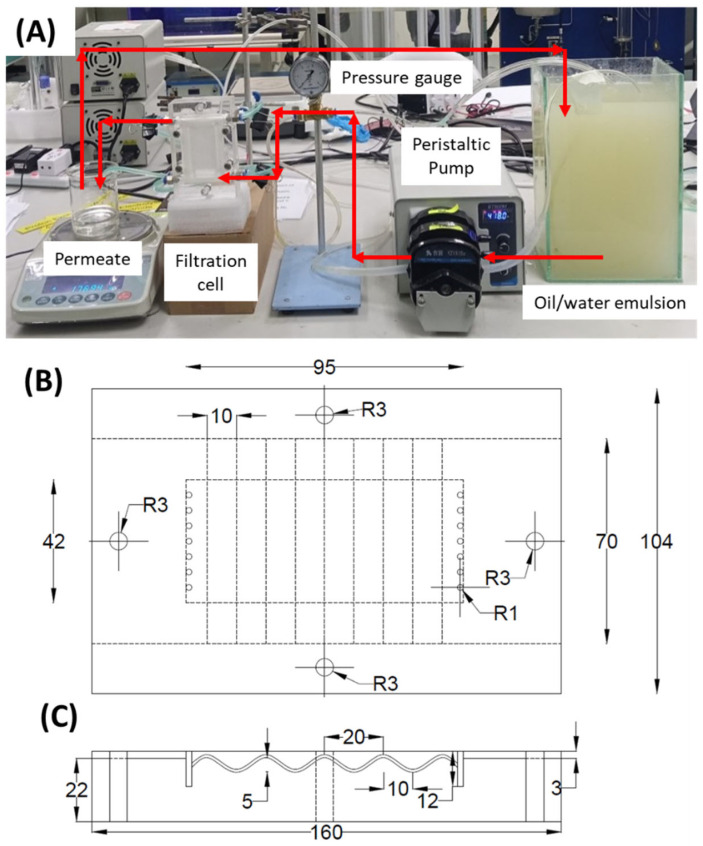
(**A**) Picture of the crossflow filtration setup, also showing the design and dimensions of the filtration cell with a wavy flow channel viewed from the top (**B**) and the side (**C**).

**Figure 2 membranes-12-01153-f002:**
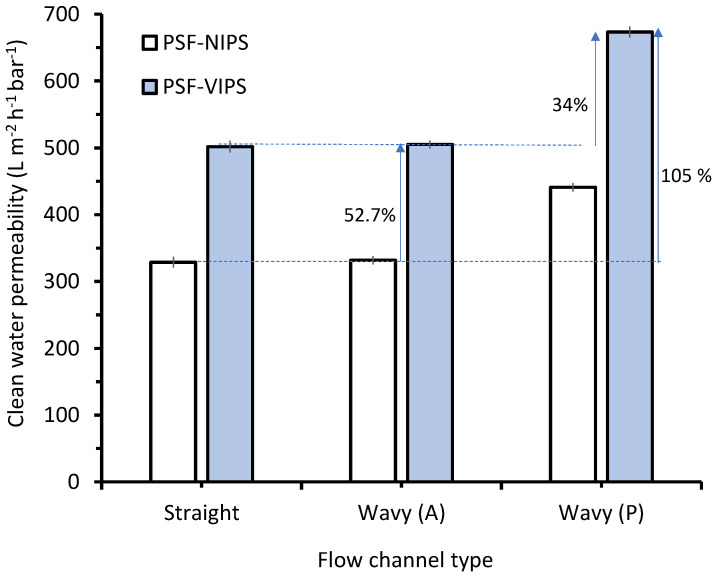
Effect of membrane material and feed flow channel geometry on clean water permeability. Two permeability calculations were done for the wavy channel. A represents the actual value and P represents the advantage of additional membrane area when projected to a straight flow channel.

**Figure 3 membranes-12-01153-f003:**
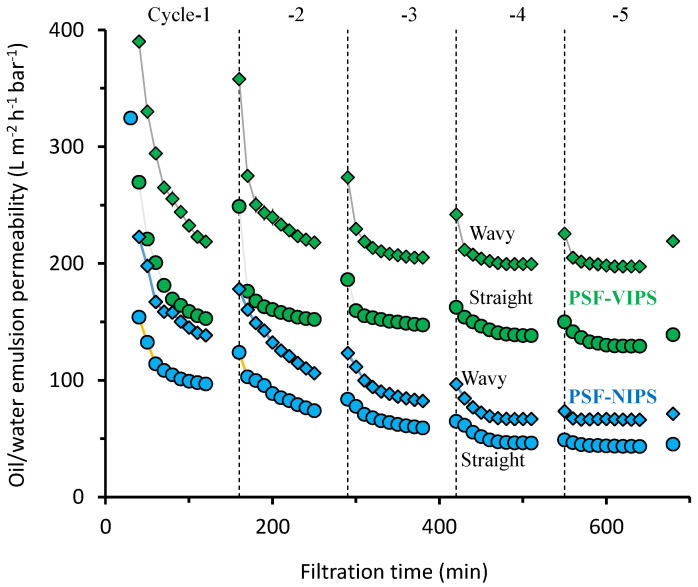
Effect of flow channel modification on membrane performance during oil/water emulsion filtration by also counting the advantage of additional surface area in the wavy flow channel.

**Figure 4 membranes-12-01153-f004:**
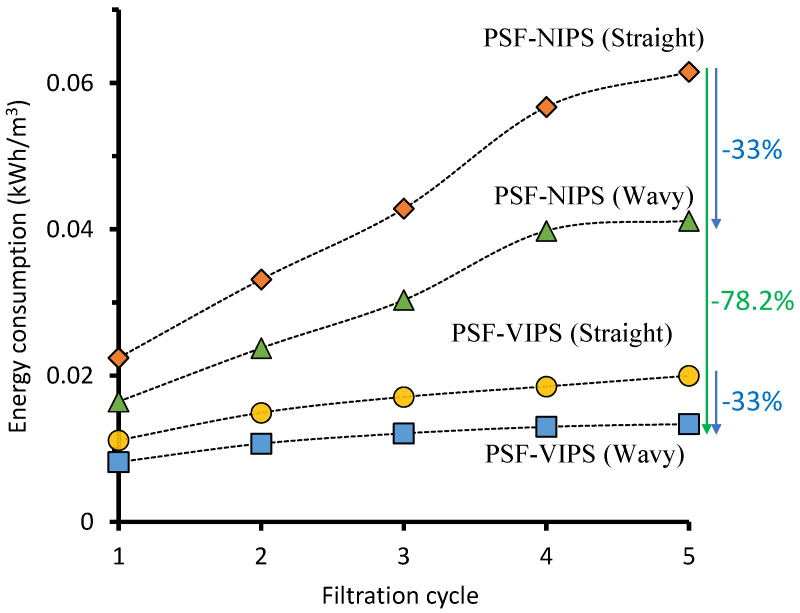
Pumping energy footprint reduction through the membrane and the flow channel development for oil/water emulsion treatment.

## Data Availability

The data presented in this study are available on request from the corresponding author.

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
