# Peer review of "The Combined Effects of the Membrane and Flow Channel Development on the Performance and Energy Footprint of Oil/Water Emulsion Filtration"

_membranes, 2022, doi:10.3390/membranes12111153_

Round 1

Reviewer 1 Report

In the present study, the author has made an attempt to address the issues of fouling during the oil/water emulsion separation. However, the same author has recently published the usage of PSF-NIPS membranes with wavy flow channels for oil/water separation (Journal of Water Process Engineering Volume 44, December 2021, 102340). In the present study, the author has used a PSF-VIPS membrane with a wavy flow channel for novelty.  At the same time, the author has not clearly mentioned what was the exact reason for the improvement of the PSF-VIPS membrane’s performance. If the authors can answer the following questions and revise the manuscript, this manuscript can be accepted.

1.     Figures 1B and C are blurred.

2.     SEM cross-sectional and top surface images of the membranes could be provided.

3.     What was the oil contact angle for both membranes?

4.     What was the reason for the PSF-VIPS membrane’s low water contact angle when compared to PSF-NIPS membranes?

5.     Although the PSF-NIPS membrane has a lower thickness than PSF-VIPS, why the water permeance is high for the PSF-VIPS membrane?

6.     How did the author evaluate the oil content in permeate and feed?

7.     The long-term performance of this membrane could be reported.

8.     The author has not compared the present study data with the literature.

9.     The size of oil droplets in the emulsion could be estimated and an optical image could be included.

10.  The digital image of feed and permeate could be given.

Reviewer 2 Report

The Manuscript ID membranes-1968992, despite being well written, does not present any novelties based on the literature. I suggest authors avoid self-citations in the development of the article.

Reviewer 3 Report

This paper reports a combined approach to tackling the membrane fouling challenge in oil/water emulsion filtration via a membrane and a flow channel. The results show that the combined membrane and flow channel developments enhanced the clean water permeability with a combined increment of 105%, of which 34% was attributed to the increased effective filtration area due to the wavy flow channel. This is a well written paper, but some details should be concerned. Overall, the work can be published in Membranes after minor revision. Some changes can be made concerning my following remarks.

1.      Line 232,the P value from ANOVA are .6316 and .6464, respectively. Please correct it.

2.      From Figure 2, it can be seen that the increasement of permeability should be attributed to the actual additional membrane area. But in Figure 3, the difference between the wave channel and the straight channel is very large. So, is the permeability actual value or the advantage of additional membrane area?

Round 2

Reviewer 1 Report

The revised version can be accepted.